# Checkpoint-Dependent Sensitivities to Nucleoside Analogues Uncover Specific Patterns of Genomic Instability

**DOI:** 10.3390/cimb47090756

**Published:** 2025-09-12

**Authors:** Zainab Burhanuddin Kagalwala, Mohammed Ayan Chhipa, Zohreh Kianfard, Essam Karam, Sirasie P. Magalage, Sarah A. Sabatinos

**Affiliations:** 1Department of Chemistry and Biology, Faculty of Science, Toronto Metropolitan University, Toronto, ON M5B 2K3, Canada; zainabhussain943@gmail.com (Z.B.K.); mohammedayan.chhipa@mail.utoronto.ca (M.A.C.); zkianfar@torontomu.ca (Z.K.); essam.karam@torontomu.ca (E.K.); smagalage@torontomu.ca (S.P.M.); 2School of Pharmacy, University College London, 29-39 Brunswick Square, Bloomsbury, London WC1N 1AX, UK; 3Yeates School of Graduate Studies, Toronto Metropolitan University, Toronto, ON M5B 2K3, Canada

**Keywords:** nucleoside analogue, synthetic lethality, fission yeast screen, half-maximal inhibitory concentration (IC50), genome instability, HIV anti-retrovirus, cell cycle checkpoint, DNA replication instability, DNA damage and repair

## Abstract

Nucleoside analogues are used as drugs and as labels in laboratory-based research. However, the effect of different nucleoside analogue mechanism(s) on cell sensitivity or mutagenesis is unclear. This is particularly important in cancer treatments where checkpoint proteins and DNA damage factors are often mutated. We tested six nucleoside analogues in fission yeast, *Schizosaccharomyces pombe*. We found that the mutations in the DNA replication checkpoint cause unique sensitivity profiles towards chemotherapeutic nucleoside analogues (gemcitabine, 5-fluorouracil, cytarabine) and the non-clinical analogue bromodeoxyuridine. Antiretroviral compounds, zidovudine and lamivudine, did not alter cell growth. We compared half-maximal inhibitory concentration (IC50) doses between checkpoint deficient yeast strains, examining culture growth and DNA mis-segregation. Intriguingly, gemcitabine and bromodeoxyuridine doses above the IC50 promoted better growth. Above each compound’s IC50 dose we saw that cells were insensitive to nucleoside analogue re-exposure, particularly in DNA replication checkpoint mutants (*cds1∆*, *rad3∆*). Thus, pairing nucleoside analogue use with personal genomics may inform drug choice, dose, and schedule. Finally, these data indicate that resistance may be predictable, informing clinical strategy.

## 1. Introduction

Nucleoside analogues mimic DNA/RNA bases. The clinical uses of nucleoside analogues range from cancer chemotherapy to the treatment of Human Immunodeficiency Virus (HIV). Nucleoside analogue drugs work by a variety of mechanisms, often causing DNA replication inhibition and/or DNA damage, e.g., [1,2,3]. Chemotherapeutics gemcitabine, cytarabine (AraC), and 5-fluorouracil (5FU) are antimetabolite prodrugs that are converted into active metabolites in vivo, and inhibit DNA replication in proliferating cells by incorporation, chain termination, and decreased dNTP levels [4]. 5FU also affects mRNA synthesis causing cell death [5,6,7,8]. Bromodeoxyuridine (BrdU) is incorporated into DNA during replication and can be detected using anti-BrdU antibodies [9]. BrdU is used to detect proliferation in laboratory studies [10,11]. Antiretroviral nucleoside analogues, zidovudine (AZT) and lamivudine (3TC), inhibit viral replication through chain termination [12] and are an important component of HIV prophylaxis and treatment regimens. In all these cases, polymerase activity and proliferation are coupled to drug efficacy.

Because of the link between nucleoside analogues and DNA metabolism, exposed cells may arrest in the cell cycle. Cell cycle checkpoints regulate transition between phases of growth, DNA synthesis, and division in response to genome instability [13]. Checkpoint loss in cancer may make cells more sensitive to various drugs. Checkpoint proteins activate regulatory cascades that promote cell cycle arrest and repair [14,15,16,17,18,19]. Checkpoints prevent DNA under-replication, chromosomal changes, and mutations. Loss of checkpoints causes genome instability and induces cycles of DNA damage, chromosome mis-segregation, and aneuploidy, e.g., [20,21,22,23,24]. Mutational changes and genomic instability have a profound impact on the development of cancer in mammals, e.g., [25,26,27,28] and development of heterogeneity within a population of cancer cells or tumour, e.g., [25,26,27]. Cancer cells frequently mutate during drug treatment and adapt, causing a therapeutic obstacle through emergent drug resistance, e.g., [29,30,31].

DNA replication arrest and DNA damage are two potent sources of genomic instability, carcinogenesis, and drug resistance [20,24,26,30,32]. DNA replication instability can be caused by a variety of environmental factors, including changes in nucleotide metabolism [33,34] or polymerase inhibition [35,36,37]. Barriers, including RNA-DNA R-loops and quadruplex DNA, may block unwinding and replication fork movement [38,39,40,41] to cause DNA damage. DNA double strand breaks (DSBs) are considered a particularly lethal form of DNA damage that activate the G2-M DNA damage checkpoint [42,43,44,45]. DSBs are caused by ionizing radiation and some drugs (e.g., phleomycin) [42,44,45,46]. A clinically useful class of DBS drugs includes camptothecin and irinotecan, molecules that bind topoisomerase to the DNA causing DNA replication fork collisions and DSBs in the next S-phase [47,48,49]. DSBs are particularly dangerous if unrepaired by anaphase, causing genome loss or gain in daughter cells, chromosome fusions, telomere loss, and chromosome instability [50,51]. Homologous recombination becomes an important downstream checkpoint response to DSBs and also DNA replication structures [52,53,54].

Some nucleoside analogues are known to cause DNA damage and mutation. Yet, the link between checkpoint kinase function (genotype), initial drug sensitivity, and emergence of resistance is not well described. Replication instability phenotypes, including single-stranded DNA accumulation, ssDNA gaps, and replication fork reversal, can cause DSBs and damage checkpoint effects in replication checkpoint mutants [55,56,57]. We used checkpoint-deficient mutants of fission yeast, *Schizosaccharomyces pombe*, to test nucleoside analogue sensitivity in either DNA replication or DNA damage checkpoint pathways. These single-gene effects can be used to deconvolve the relationships between dose and resistance. *S. pombe* is a rod-shaped, unicellular eukaryote that is phylogenetically distinct from the budding yeast *Saccharomyces cerevisiae* [58,59]. Nucleotide metabolism, chromosome structures, and some DNA damage and repair mechanisms are more similar between *S. pombe* and humans than in budding yeast [10,58,60]. We tested single-gene loss of function mutations for nucleoside analogue effects, to compare with loss of checkpoint in human disease. Single mutations are stringently defined in our fission yeast model, which is not the case in mammalian cell lines that undergo mutations during transformation.

We developed a rapid proliferative assay to detect *S. pombe* culture proliferation and survival [61]. These results directly support nucleoside analogue studies in fission yeast performed in our group (Kianfard et al., in preparation; and [10]) and others [11,62,63,64,65]. We found that anticancer nucleoside analogues are most affected by loss of the DNA replication checkpoint (*cds1∆*, *rad3∆* mutants). Loss of the Rad3 apical checkpoint kinase (ATR orthologue; *rad3∆* mutant) causes death at the half-maximal inhibitory concentration (IC50) dose. We propose that the IC50 dose is a “crisis point”, above which cell growth becomes less affected and potential for drug resistance increases. Antiretroviral drugs do not affect cell proliferation; thus, AZT and 3TC do not induce a catastrophic checkpoint response. We hypothesise that increased sensitivity to nucleoside analogues in checkpoint mutant cells triggers DNA mis-segregation in replication checkpoint mutant cells (*cds1∆*, *rad3∆*). We conclude that nucleoside analogue chemotherapy treatments should be considered in the context of personal genomics, and that nucleoside analogues should not be paired with ATR checkpoint kinase inhibitors clinically.

## 2. Materials and Methods

### 2.1. Yeast Strains

Fission yeast cells were cultured following standard conditions, e.g., [66,67]. Cells were grown from frozen stocks on yeast extract with supplements (adenine, histidine, leucine, uracil, lysine; YES 225 formulation, Sunrise Bioscience, San Diego, CA, USA) at 30 °C. Cultures for nucleoside analogue exposure were incubated at 30 °C overnight in pombe minimal glutamate (PMG) media supplemented with uracil and adenine (both 225 mg/L; BioShop, Burlington, ON, Canada). Other supplements were added as appropriate at 225 mg/L. Strain genotypes are listed in Table 1.

*Schizosaccharomyces pombe* lacks a thymidine salvage pathway. Strains used to assess nucleoside analogue sensitivity contained Herpes Simplex Virus 1 (HSV1) thymidine kinase that was stably integrated and expressed from a constitutive alcohol dehydrogenase promoter (*Padh1*-*hsv-tk^+^*). The human equilibrative nucleoside transporter 1 (hENT1) was stably integrated into strains using the *Padh1* promoter (*Padh1*-*hENT1^+^*). Expression of hENT1 allows better nucleoside analogue uptake at lower doses, as described in [10,65].

### 2.2. Drugs

Nucleoside analogue drugs were resuspended in water or DMSO as stock solutions of 100 µM. The drugs used include gemcitabine (Toronto Research Chemicals, Toronto, ON, Canada), cytarabine (AraC; SigmaAldrich, Mississauga, ON, Canada), 5-fluorouracil (5-FU; BioShop Canada), 5-fluoro-2′-deoxyuridine (FUdR; BioShop Canada), bromodeoxyuridine (BrdU; SigmaAldrich Canada), and thymidine (BioShop, Canada). Antiretroviral compounds zidovudine (AZT) and lamivudine (3TC) were kind donations from Dr. Russell Viirre (Toronto Metropolitan University, Toronto, ON, Canada). Canavanine sulfate (Cedarlane, Burlington, ON, Canada) was added to PMG + HULA medium (Appendix A) at 70 µg/mL [68].

### 2.3. Proliferation Assay and IC50

Liquid culture growth was used to evaluate the dependency of nucleoside analogue response on the cell cycle checkpoint. Decreased proliferation in the drug could indicate cell death or arrest. *S. pombe* checkpoint mutant cells (Table 1) were grown overnight in pombe minimal glutamate (PMG) media supplemented with 225 mg/mL uracil and adenine. Cultures were diluted in fresh PMG + UA to an OD600 of approximately 0.1 Diluted cultures were kept at room temperature for 2 h to acclimate. Well plates were prepared containing 200 µL of diluted cells, PMG + UA medium, and drug. Each drug was prepared and used at final concentrations of 0, 0.05, 0.25, 0.5, 2.5, 5.0, 25, 50, 125, 250, 375, and 500 µM; these doses generate an 11-point IC50 curve.

Optical density measurements at a wavelength of 600 nm (OD600) were taken using a spectrophotometer in 96-well growth plates. OD600 readings were taken before static incubation at 30 °C, and after 24 h and 48 h of growth at 30 °C. After the 48 h OD600 reading, 96-well plates were spotted onto yeast extract with supplements (YES; Sunrise Media, Carlsbad, CA, USA) and solid media (1.8% agar; BioShop), prepared and autoclaved as described. Spotting was performed using a 48-array pinner (V&P Scientific, San Diego, CA, USA, 3 µL hanging droplet). Pinned plates were then incubated at 30 °C overnight. The following day, the YES plates were scanned for analysis. Cells were also spotted onto PMG + HULA containing either FUdR or 70 µg/mL canavanine with Phloxine B. Spots were incubated for 3 days (FUdR) or 9 days (canavanine) before assessment.

Aggregate measurements were calculated from 4 replicate experiments. Heat maps were constructed in GraphPad Prism10. For checkpoint mutant sensitivity assessment, the average IC50 and toxic dose values from aggregate values were binned with a 4-point scale: 0 weak effect—500 µM or higher; 1—200 to 500 µM; 2—50 to 200 µM; 3—<50 µM, strong effect. The 4-point scale was used to create a checkpoint dependency prediction heat map for DNA replication checkpoint effects (strongly in *rad3∆* and *cds1∆*) and DNA damage checkpoint effects (*chk1∆* and *rad3∆* affected).

### 2.4. Microscopy

Cells were fixed in cold 100% ethanol to a final concentration of 70% ethanol. Samples were vortexed and stored at 4 °C. To stain, fixed cells were rehydrated in water and then incubated in aniline methylene blue solution (AMB; 1% aniline blue water-soluble powder *w*/*v*, in 1 × PBS) for 15 min at room temperature. AMB binds to the *S. pombe* septum and fluoresces to specifically reveal septa. AMB-stained cells were smeared onto glass slides and air dried at room temperature. Samples were mounted in 4 μL of fluorescent imaging mounting medium (50% glycerol, 1% DABCO) containing 1 μg/mL DAPI. Samples were sealed with a coverslip and stored in a light-protected slide box at −20 °C. Cells were imaged on an Olympus IX83 Inverted Microscope (Olympus Corporation, Tokyo, Japan) linked to a Hamamatsu ORCA-Flash 4.0 CMOS digital camera (Hamamatsu Photonics K.K, Hamamatsu, Japan). CellSens acquisition software (Dimension 1.13 Olympus Corporation, Tokyo, Japan) was used to acquire images on DAPI-excitation 359 nm/emission 461 nm, and bright field settings. FIJI software was used to analyse the images [69] version 2.3.0/1.53f.

### 2.5. Statistical Analysis

IC50 OD600 values were imported into GraphPad Prism 10 for analysis. Drug concentrations were log10-transformed. A 3-parameter, non-linear fit of dose–response was fit to the data. IC50 values were calculated for each independent replicate. Graphs were made using the 48 h OD600 readings for comparison with calculated IC50 values. Replicate IC50 values were averaged and assessed using standard deviation between calculated values. Microscopy images were examined, and cells were measured tip-to-tip or tip-to-septum. A pairwise two-tailed *t*-test was used to compare cell lengths between treatment conditions and genotypes (GraphPad Prism 10).

## 3. Results and Discussion

*Schizosaccharomyces pombe* replication and damage checkpoints are initiated by the Rad3 kinase (Figure 1) [16,70,71]. Cells lacking *cds1*∆ or *rad3∆* cannot activate the Intra-S and S-G2 checkpoints during replication stress; this causes DNA damage. Because the DNA replication checkpoint is important in processing nucleoside analogue effects, *cds1∆* cells elongate, accumulate DBSs, and enter a lethal G2 arrest like in hydroxyurea [16,17,72]. Conversely, cells lacking *chk1∆* can arrest in S-phase and stabilize DNA replication. However, *chk1∆* cells cannot arrest in G2 if there is DNA damage [17,18,70]. Thus, *chk1∆* cultures are a test for DSB-specific effects of nucleoside analogues. Because *rad3*∆ cannot arrest in either DNA replication instability or damage, we predicted that *rad3∆* cells are most sensitive to nucleoside analogue effects. We expected that *rad3∆* cells would present with DNA mis-segregation and “cell untimely torn” (*cut*) divisions, in which the septum forms over entangled DNA [73].

We tested thymidine- and cytidine-like nucleoside analogues (Figure 1) in *S. pombe* strains containing Herpes Simplex Virus (HSV1) thymidine kinase (*hsv-tk^+^*) and human equilibrative nucleoside transporter (hENT1^+^) transgenes (Table 1). Since *hsv-tk^+^* is a thymidine kinase, we hypothesized that thymidine-analogues would show the greatest effects. Because excess thymidine alters dNTP pools and causes S-phase arrest in mammalian cells [74] and fission yeast [10], we predicted that all thymidine analogues would arrest cell growth. BrdU causes mutation and cell death in *S. pombe* DNA replication checkpoint mutants [10]. We expected lower IC50 values for these *cds1∆* and *rad3∆* cells.

### 3.1. Thymidine Analogues Induce Both DNA Replication and DNA Damage Stress

We used our *S. pombe*-adapted dose–response methods to examine culture growth by OD600 and to calculate half-maximal inhibitory (IC50) doses for each analogue [61]. IC50 values allow comparisons between strain sensitivities and different drugs [61,75]. IC50 values are used to suggest a drug dose that decreases proliferation [75]. Because *S. pombe* cells elongate during G2-checkpoint arrest, OD600 may increase simply because of bigger cells leading to higher turbidity. To determine if cells were truly dead or merely arrested, we spotted cultures onto drug-free medium after exposure.

We found that thymidine, 5 fluorouracil (5FU), and bromodeoxyuridine (BrdU) all decreased *cds1∆* and *rad3∆* mutant cell growth, while zidovudine (AZT) did not. Thymidine IC50 values were 16.8 µM for *cds1∆* and 14.9 µM for *rad3∆*; however, thymidine did not decrease OD600 or generate an IC50 value in wild type or *chk1∆.* A difference in growth was reflected in area under the curve (AUC), which was significantly lower in *rad3∆* than wild type. The *rad3∆* spot growth above 5 µM thymidine was slow with fewer cells per spot (Figure 2(Aiv)). These data confirm that high levels of thymidine are toxic to *rad3∆* cells. Cultures of *chk1∆* cells had less spot growth at the highest thymidine doses. The fact that *cds1∆* cells show similar IC50 curves in liquid growth indicate that thymidine causes DNA replication instability that relies on the DNA replication checkpoint for stability. However, because *rad3∆* cells are more sensitive than *cds1∆*, we infer that thymidine effects are not solely caused by replication checkpoint loss. Instead, *chk1∆* growth reduction in spots after exposure to >100 µM thymidine indicates that high-dose thymidine causes genome instability, possibly through DNA damage.

In contrast, 5-fluorouracil (5FU) and bromodeoxyuridine (BrdU) affected the growth of all strains. 5FU was most potent in *rad3∆* and *cds1∆* (IC50s of 34.5 µM and 72.0 µM, respectively) but 10-fold more 5FU is required to cause growth effects in wild type and *chk1∆* (IC50s of 388 µM and 643 µM). Spot growth after 48 h of 5FU exposure is minimal in *rad3∆* and *cds1∆* above 5 µM (Figure 2B). Wild type and *chk1∆* cells have reduced spot growth > 50 µM. In all cases, the concentrations at which we observed reduced spot growth (Figure 2(Biv)) were lower than the calculated IC50s.

These data indicate that 5FU effects require the DNA replication checkpoint to preserve viability. However, even replication-checkpoint competent cells are sensitive to high doses of 5FU, which impairs cell growth. This may reflect that the primary mechanism of 5FU is reportedly upon RNA synthesis [5,6,7,8]; consequently, all our mutants tested would be susceptible to 5FU effects. While *chk1∆* growth is similar to wild type after 5FU, the subtle difference in IC50 and maximum unaffected doses in spots argues that impact of DNA damage checkpoint loss should be investigated for differences in mutation kind and profile (i.e., [76]).

We found that BrdU also decreased OD600 and spot growth in all strains, and at lower doses than 5FU. BrdU is an important molecular tool to monitor DNA synthesis. Yet, the potential BrdU effects on DNA damage and proliferation are frequently overlooked in the interest of the desired experimental goal. BrdU IC50 values ranged from 2.8 µM in *rad3∆* to 11.1 µM in wild type (Figure 2(Ci,Cii)). AUC for *rad3∆* was significantly different from wild type (*p* < 0.05, Figure 2(Ciii)). While all spots showed diminished growth above 2.5 µM BrdU, *cds1∆* spots were less affected by BrdU treatment (Figure 2(Civ)). Intriguingly, *chk1∆* cells showed less growth by IC50 (5.4 µM) and in spots compared to *cds1∆.* These data indicate that BrdU generates substantial DNA damage that may be more impactful than its effect on DNA replication stability. Because *cds1∆* cells grew better than wild type after BrdU treatment, we conclude that the loss of the DNA replication checkpoint in *cds1∆* but presence of the DNA damage checkpoint may improve survival. In contrast, we infer that although OD600 turbidity is decreased in *rad3∆* and *chk1∆*, these cells acquire DNA damage and do not grow in the presence of BrdU. After spotting onto BrdU-free medium to grow post-exposure, the *rad3∆* and *chk1∆* cells re-enter growth slowly. We conclude that slower *rad3∆* or *chk1∆* growth in BrdU is caused by DNA damage; these G2/M checkpoint-deficient strains cannot repair or arrest in the presence of BrdU-induced DSBs.

AZT is an antiretroviral polymerase inhibitor that did not inhibit growth in culture (Figure 2(Di)). All strains generated relatively flat dose–growth curves that cannot be used to calculate IC50 values. We reported all IC50 values as >500 µM, the highest dose that we tested. No significant difference in AUC values or spot growth was observed. Cells after 48 h of AZT did not die but grew into full spots, even at 500 µM. We conclude that AZT is a non-toxic thymidine nucleoside analogue that does not impact growth during or after treatment, nor is this dependent on the DNA replication or damage checkpoints. This contrasts with 5FU, BrdU, and thymidine, all which arrest cell growth and cause long-term effects that decrease growth after exposure.

### 3.2. Cytidine Anticancer Analogues Generate Replication Instability and Not DNA Damage Arrest

Cytidine-related nucleoside analogue drugs include gemcitabine (Gem), cytarabine (AraC), and the antiretroviral lamivudine (3TC). We used the same strains containing HSV1-TK. HSV1-TK is a promiscuous kinase that can phosphorylate a broad array of nucleosides including cytidine [38,40]. Because altered CTP and dCTP levels result in mutagenesis, we hypothesized that HSV1-TK may cause genome instability effects from cytidine nucleoside analogue drugs. If so, cytidine-dependent DNA replication instability and/or DNA damage would be detected in specific checkpoint mutants. We predicted that replication checkpoint kinase mutant *cds1*∆, or apical kinase *rad3∆*, would show the least growth in cytidine analogues in our transgenic strains expressing hENT1^+^ and HSV1-TK.

We found that gemcitabine and AraC both decrease the growth of replication checkpoint mutant cells. In this fission yeast-IC50 test of comparative dose–response effects, gemcitabine caused growth arrest in *cds1*∆ and *rad3∆* cells (Figure 3A, IC50 values of 2.7 µM and 0.9 µM, respectively). The area under the curve for *cds1∆* and *rad3∆* was also significantly lower than wild type cells in gemcitabine. Both *cds1∆* and *rad3∆* cells had decreased spot growth above 50 µM doses. Oddly, *cds1∆* and *rad3∆* had improved growth at the highest gemcitabine doses. We hypothesized that high-dose gemcitabine promotes drug resistance, particularly in *rad3∆* cells. Large colonies also emerged at the highest doses in *cds1*∆ cells after gemcitabine exposure and outgrowth. Future work will determine whether these large colonies harbour suppressor mutations caused by gemcitabine exposure. Together, our data suggest that gemcitabine effects on growth and mutation were highest in replication checkpoint mutants *rad3∆* and *cds1*∆, and less effective at high dose.

Gemcitabine did not stop culture growth in wild type cells. The wild type dose–response curve, AUC values, and spot formation were all consistent with a conclusion that gemcitabine does not inhibit growth or kill cells in doses up to 500 µM. We recorded the IC50 value of gemcitabine in wild type as >500 µM from multiple replicate IC50 calculations that were averaged (Table 2). Intriguingly, when IC50 was calculated from normalized and aggregated data in the chart of Figure 3, *chk1*∆ cells generated an IC50 value of 2.8 µM gemcitabine (Figure 3(Aii)). Yet, the dose–response curve is not classically sigmoidal, and the degree of inhibition by gemcitabine on *chk1∆* is low (Figure 3(Aiii)—lowest growth is 0.8 of normalized OD600). The area of *chk1∆* curves in gemcitabine was not significantly different from wild type. The *chk1∆* spots grew after all exposure doses. These data indicate that gemcitabine is not toxic to wild type or *chk1∆* in the conditions that we tested.

In contrast, cytarabine (AraC) decreased growth of all checkpoint mutant cultures (Figure 3B). The *rad3∆* cells had an IC50 of 0.15 µM and a significantly lower AUC compared to wild type (IC50 AraC > 500 µM). Growth of *rad3∆* cells after AraC was tested by pinning onto plain medium, and *rad3∆* cell growth was minimal or non-existent following even the lowest dose of 0.05 µM AraC and up to 250 µM. At the two highest doses, *rad3∆* cells show some growth post-exposure (Figure 3(Biv)). We hypothesize that high-dose AraC limits toxicity at the highest doses in *rad3∆*. The *cds1∆* cells had an IC50 of 13.1 µM and do not form spots above a dose of ~5 µM. The *chk1∆* cells had an IC50 of 24.5 µM, and do not grow above the IC50 dose of ~25 µM. In the *cds1∆* and *chk1∆* samples, toxicity caused by AraC is coincident with the IC50 dose. OD600 values of non-cds1∆ cells did not capture the effect of AraC on replication-checkpoint competent cells. Replication-checkpoint mutants (i.e., *cds1*∆) may therefore experience arrest caused by DNA damage more quickly. We propose that DNA damage induction in *cds1∆* inhibits growth and affects OD600 and thus IC50. However, replication-checkpoint competent cells (i.e., wild type, *chk1*∆) are damaged by AraC. In *chk1∆*, the effect is lethal above 25 µM.

We tested the antiretroviral drug lamivudine (3TC, Figure 3C) to determine how a cytidine-analogue viral polymerase inhibitor was tolerated in *S. pombe* mutants. In the IC50 test, all cultures grew and an IC50 dose could not be calculated (>500 µM). AUC values were similar for all strains. However, we repeatedly saw that checkpoint mutants had reduced and slower spot growth at the highest doses of 3TC (Figure 3(Civ)). This spot outgrowth indicated that although 3TC is generally non-toxic, there may be some dependency on checkpoint functions to survive high 3TC doses.

### 3.3. IC50 Doses of BrdU and Gemcitabine Induce DNA Mis-Segregation, Whilst Higher Doses Promote Drug-Resistant Morphologies

During IC50 assessment, we saw that 5FU, AraC, gemcitabine, and BrdU sometimes caused more growth in spots post-treatment and above the IC50 dose. We compared drug doses between each strain over 48 h, and then spotted for viability on the same plate to compare growth differences due to each drug within a strain (Figure 4A). Replication checkpoint mutant strains *cds1∆* and *rad3∆* showed a decrease in OD600 in nucleoside analogue around the IC50 dose. Yet, above the IC50 dose *cds1∆* and *rad3∆* cells often grew large and atypical colonies. Reminiscent of “suppressor” mutant colonies, these large colonies grew faster than surrounding cells on the spot background area above the IC50 dose. 

The highest doses frequently allowed spot growth that resembled sub-IC50 treatment. Wild type and *chk1∆* cultures rarely showed this pattern of more growth and suppressors higher doses. Our data suggested that nucleoside analogue IC50 exposure may cause a genomic instability “crisis” in replication-checkpoint defective cells. This IC50 crisis and outgrowth presented a potential model to explore drug resistance from nucleoside analogue exposure.

Because our data suggested that the maximally effective dose for a specific mutant strain–nucleoside analogue pairing is approximately the IC50 value, we predicted that cells exposed to an IC50 dose are more likely to die and show obvious signs of genome instability. In this hypothesis, most cells in the IC50-dose-treated culture die and cannot grow into spots. To compare morphology with spot results (Figure 4A), we used DNA and aniline blue septum staining [77,78]. We imaged cells and compared morphologies under three drug conditions: no drug, the IC50 dose, and the highest dose at 500 µM. We began with wild type and *rad3*∆ cells in gemcitabine because of the stark difference in IC50 response. Wild type and *chk1∆* cultures did not generate a sigmoidal curve or IC50 value in gemcitabine using OD600; we inferred that DNA damage response is not essential to stopping growth. Because wild type cells were insensitive to gemcitabine and did not generate an IC50 value, lower-dose IC50-dose cells could be tested.

However, this did not mean that wild type cells were unaffected by gemcitabine treatment. We found that wild type cells in 500 µM gemcitabine were long and multi-septated, with clear DNA mis-segregation (Figure 4B). We compared wild type cell length in untreated and 500 µM gemcitabine and found that gemcitabine treatment caused cell elongation (Figure 4C). The number of anaphase and septated wild type cells was higher in 500 µM gemcitabine, showing that G1-S phase arrest occurs (Figure 4D). Nuclear mis-segregation was increased in wild type after 500 µM gemcitabine (Figure 4E). We conclude that wild type cells are sensitive to high-dose gemcitabine (Figure 3(Ai), black line), reflecting gemcitabine sensitivity, even though no lower-dose IC50 is calculated. Further, we infer that wild type DNA damage checkpoint prevents division in gemcitabine, correlated with cell elongation and septation phenotypes.

In contrast, *rad3∆* cells had a gemcitabine IC50 value of approximately 1 µM (Table 2). *rad3∆* cells at the IC50 dose showed multiple nuclei, *cut* DNA, elongation, and multiple septa (Figure 4B). This was consistent with IC50 doses of gemcitabine causing genomic instability in *rad3∆* cells. At the highest 500 µM dose of gemcitabine, *rad3∆* cells were more uniform in size, although longer than untreated (Figure 4C, Table 3); we attribute some of this difference to cell growth arrest in untreated cultures after 48 h static growth. More *rad3∆* cells had septa or anaphase nuclei in either IC50 or 500 µM gemcitabine, indicating more cells were in G1-S phases (Figure 4E). Morphologically, *rad3∆* cells in IC50 gemcitabine looked like wild type cells at the highest (500 µM) dose (Figure 4B). By counting abnormal nuclear segregation of DAPI, we found that *rad3∆* cells at 500 µM gemcitabine had fewer obvious DNA mis-segregation events than at the IC50 dose (Figure 4D). Our data indicate that *rad3∆* cells are most sensitive to gemcitabine at the IC50 dose where genome instability is obvious.

Because 500 µM gemcitabine doses impacted *rad3∆* cells less, we hypothesized that other nucleoside analogues may arrest replication checkpoint mutant cells, but not kill at 500 µM doses. We tested BrdU, because all strains were sensitive to BrdU, and we calculated IC50 doses (Figure 2C, Table 2). We predicted that *cds1∆* or *rad3∆* cells at an IC50 dose would show more DNA mis-segregation and abnormal morphologies (i.e., longer cells or septated/anaphase cell accumulation) than at the highest dose. Instead, we found that all genotypes showed dramatic changes in morphology at their respective BrdU IC50 doses (Figure 5A), including elongation, abnormal septation, aneuploidy, and multinucleated cells. Intriguingly, cells that were treated with 500 µM BrdU had morphologies like untreated and cycling fission yeast.

*Schizosaccharomyces pombe* cells elongate during G2 arrest and in response to DNA damage and may show increased DNA mis-segregation as a symptom of chromosome instability. Elongation could also contribute to higher OD600 measurements, which might affect growth curves. To test for elongation, we measured cell lengths after IC50 and high-dose BrdU. We found that BrdU at low- or high-dose caused longer cells and more variation (Figure 5B). Wild type cells elongated in IC50 doses and remained longer at high dose (Table 4); this implies that there was a G2-arrest or block to cytokinesis. In contrast, checkpoint mutants showed longer cells at the IC50 dose, the populations were generally shorter at 500 µM (Table 4). BrdU increased the proportion of binucleate and septated cells at each dose (Figure 5C), indicating G1/S accumulation and arrest. DNA mis-segregation increased in all cells and at either IC50 or 500 µM doses. However, the IC50 dose of BrdU caused the highest frequency of mis-segregation in all cells (Figure 5D), showing that the dose of most sensitivity is correlated with the largest biological effect. Checkpoint mutants had less mis-segregation at 500 µM BrdU, with no significant difference between 500 µM and untreated.

Finally, we tested whether cells were still sensitive to nucleoside analogue treatment after BrdU (Figure 5E, Appendix A). We spotted cultures onto 5-fluoro-2′-deoxyuridine (FUdR) to assess HSV-TK function. FUdR added to the medium detects the emergence of nucleoside analogue-resistant cells, because the analogue can no longer be phosphorylated [10,11]. We saw that wild type, *cds1∆*, and *rad3∆* cells had few FUdR-resistant cells after incubation without BrdU. In contrast, *chk1∆* cells have a higher frequency of FUdR resistance that may indicate general genome instability and HSV-TK loss even without BrdU. With increased BrdU dose the number of FUdR-resistant colonies within each spot increased. This occurred even as the overall number of cells per dot decreased with increasing FUdR doses (Figure 2(Civ)). We spotted onto canavanine to assess forward mutation of canavanine sensitivity. We found that *cds1∆* and *chk1∆* cells have more FUdR growth in lower-concentration spots, up to the IC50. Contrastingly, the *rad3∆* cells have FUdR-resistant colonies that form in doses up to the IC50, and again at the highest doses of BrdU (Appendix A).

We conclude that lower-dose BrdU promotes DNA replication arrest and DNA damage induction (Figure 6). Generally, we found that an IC50 BrdU dose is often more toxic to cells (Figure 6A,B,D). These nucleoside analogue effects are distinctive from those caused by hydroxyurea (HU; Appendix A). HU depletes dNTP levels and causes DNA replication arrest through the S-phase checkpoints. Consequently, the IC50 values of HU are lowest in *rad3∆* and *cds1∆* (0.74 µM and 1.1 µM, respectively, in PMG medium), followed by *chk1∆* (7.6 µM), and highest for wild type (14.6 µM). We saw that medium choice affects HU sensitivity (Appendix A). These HU results indicate that gemcitabine, BrdU, and AraC sensitivities in our system were more HU-like and replication-stress dependent. However, it is difficult to deconvolve effects of the analogues on ribonucleotide reductase inhibition versus direct polymerase inhibition, such as the masked chain termination of gemcitabine [79] or DNA damage induction of either BrdU or gemcitabine [80,81,82]. Further, the impact of nucleoside analogues on mRNA metabolism may impact DSBs, mutation, and genome stability, as reported for 5FU [5,8].

Intriguingly, high-dose BrdU above the IC50 dose generates a different spectrum of damage and/or mutation. Our FUdR tests show that the changes caused by BrdU may decrease nucleoside analogue sensitivity through altered HSV-TK function; this may promote cell survival and drug resistance. Because lower DNA mis-segregation was observed at the highest doses of BrdU (Figure 5D and Figure 6D), we propose that high-dose BrdU induces changes that promote survival, decrease nucleoside analogue toxic effects, and may promote downstream drug resistance.

## 4. Conclusions

Using a rapid synthetic biology method for *S. pombe*, we show that nucleoside analogue drug-dose outcome is influenced by cellular DNA replication (Cds1-dependent) or DSB repair (Chk1-dependent) checkpoints, consistent with the previous literature regarding other models and methods [63,81,83,84]. Thus, fission yeast liquid growth screens using OD600 can identify drug mechanisms [62] and mutational outcomes [84]. Capitalizing on the similarities of *S. pombe* and human checkpoint proteins and dNTP metabolism pathways [10,58], we show that the IC50 dose for cytotoxic nucleoside analogues is linked to checkpoint competency. IC50 is lowest with loss of the ATR-homologue Rad3. Drug mechanisms specific to each analogue may influence IC50 dose, treatment failure, or risk of resistance. An IC50 dose of nucleoside analogue causes DNA mis-segregation and G1/S-phase arrest. The relationship between genetic mutation background and the IC50 dose required to reduce cell growth (Figure 2 and Figure 3; Table 2) maximizes cell killing without off-target effects.

BrdU and gemcitabine doses above a strain specific IC50 dose allowed growth recovery and decreased FUdR sensitivity (Figure 4 and Figure 5). FUdR insensitivity implies that HSV-TK function was reduced. Doses above the IC50 dose also caused large colonies suggestive of suppressor mutations, and fewer morphological changes in cell length or DNA mis-segregation. Thus, nucleoside analogue doses that are too high may increase unanticipated drug insensitivity and risk of later resistance. Because gemcitabine sensitivity depends on checkpoint competency and growth medium [62,63,84], failing to consider genomics and outcome relative to dose may lead to incorrect dosing, drug insensitivity, and mutational changes within a once-sensitive population of cells.

A predictable relationship between a specific nucleoside analogue, an individual’s cellular genotype or phenotype, drug sensitivity range, and loss of sensitivity threshold may allow decreased drug use with synthetic lethality principles. Synthetic lethality uses two conditions that are non-lethal independently to cause better cell killing when both are combined [85,86]. In cancer, a synthetic lethality strategy could target tumour cells for nucleoside analogue effects, maximizing tumour death, minimizing risk of resistance, and sparing non-cancerous “bystander” cells. Homologous recombination pathway testing in this system may help to minimize the onset of mutations and drug resistance. This is similar to the strategic lethality principle of the drug Olaparib, e.g., [87,88,89]. Olaparib inhibits polyADP ribose polymerase (PARP), blocking repair and enhancing cell death in BRCA1- or BRCA2-deficient tumours [86,87,88,89].

Our method can be applied to find a specific IC50 dose that reduces cell growth (Figure 2 and Figure 3; Table 2) while minimizing the risk of failure or resistance above the IC50. This could be part of a path to uncover new genetic factors that increase nucleoside analogue efficacy. Testing for suppressor mutations that develop at the IC50 dose in checkpoint mutant cells (i.e., *cds1∆*, *rad3∆*) may improve treatments and predict causes of drug resistance. To this end, the emergence of suppressor-like colonies in 5-FU, BrdU, AraC, or gemcitabine treatment may reflect stable isolates that are less sensitive to subsequent doses of the drug. However, we see off-target effects in *rad3∆* above the IC50 dose and loss of FUdR sensitivity. Because ATR is the homologue of *S. pombe* Rad3, the use of ATR inhibitors to increase nucleoside analogue effects (i.e., [90,91]) could confer a risk of mutation and later drug resistance. ATR inhibition may allow lower clinical drug doses; future work will determine whether these lower doses are due to enhanced cell killing or if there is a higher risk of analogue resistance.

Genotype–dose calculation for patients supports their quality of life during chronic HIV prophylaxis and cancer therapy. Because nucleoside analogues target proliferating cells such as hair follicles, intestinal epithelium, and bone marrow, drug side effects can include hair loss, nausea, anemia, and compromised immunity. Even antiretroviral drugs such as AZT and 3TC can cause these effects, and also lipodystrophy and metabolic diseases (e.g., hyperglycemia, hypercholesterolemia) [92,93]. AZT and 3TC do not typically target DNA polymerase, kill uninfected cells, or stop proliferation; our data indicate that higher 3TC doses change *rad3∆* spot growth supports effects that are subtle and more mutation-based over long-term treatment [94]. Patient side-effects can cause therapy refusal leading to co-morbidities, disease progression, and accelerated drains on patient–caregiver–health networks, even if the drug is effective [95]. Using the lowest possible dose improves quality of life and may extend the window of specific nucleoside analogue use before a switch is required due to acquired drug resistance.

We conclude that personal genomics profiling ahead of treatment may predict outcomes. *Schizosaccharomyces pombe* screening using these methods could be used to amplify nucleoside analogue sensitivities and minimize the risk of resistance during long-term use. The *S. pombe wee1∆* mutant led to the discovery of Wee1 inhibitor Adavosertib. Adavosertib is now in clinical trials targeting uterine, lung, ovarian, renal and refractory tumours [76,96,97]. Wee1 inhibition can synergistically enhance radiation and gemcitabine treatment in cancer [96,98,99,100]. Thus, targeting inhibitors of cell cycle and checkpoint kinases may improve nucleoside analogue drug use through synthetic lethality, while avoiding mutagenesis and risk of later insensitivity.

## Figures and Tables

**Figure 1 cimb-47-00756-f001:**
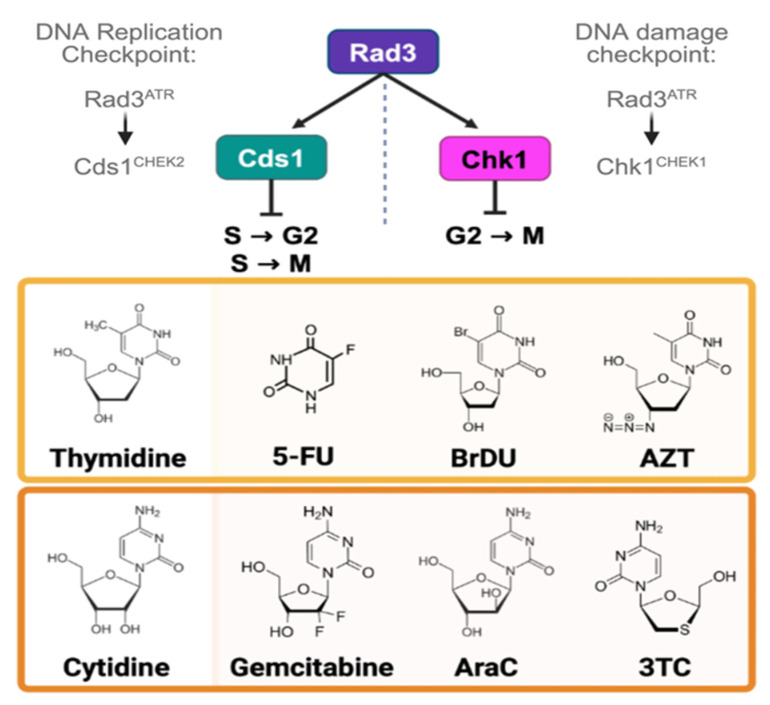
Checkpoint pathways and nucleoside analogues. A flow chart of the DNA replication (left) and DNA damage checkpoint pathways in *S. pombe*. Rad3, homologue of human ATR, is an upstream kinase that is activated in response to both DNA replication instability and DNA damage, primarily DNA double strand breaks (DSBs). During replication stress, Rad3 phosporylates Cds1 kinase, homologue of human CHEK2. In G2 response to DSBs, Rad3 phosphorylates Chk1 kinase, homologue of human CHEK1. Below, chemical structures of nucleoside analogues are shown. In the middle, thymidine is compared to its analogues 5′fluorouracil (5′FU), bromodeoxyuridine (BrdU), and zidovudine (AZT). The bottom row shows a chemical structure of cytidine and its analogues gemcitabine, cytarabine (AraC), and lamivudine (3TC). Created in BioRender. Sabatinos, S. (2025) https://BioRender.com/39ftplw (accessed on 10 August 2025).

**Figure 2 cimb-47-00756-f002:**
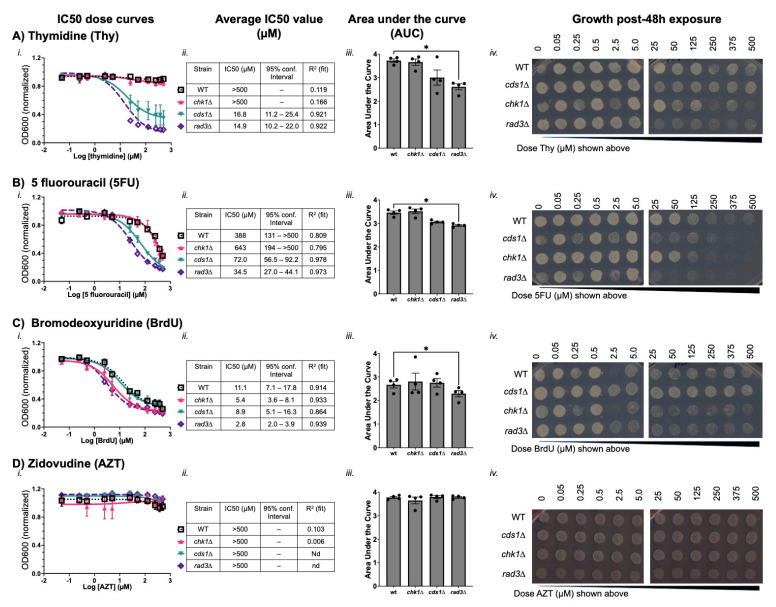
DNA replication checkpoint activity protects cells from thymidine analogue toxicity in fission yeast. (**A**–**D**). Dose–response curves of *S. pombe* liquid culture growth. Optical density at 600 nm was used to measure proliferation. Shown are average points/curves with standard deviation from a minimum of 4 experimental replicates in (**A**) thymidine (Thy), (**B**) 5 fluorouracil (5FU), (**C**) bromodeoxyuridine (BrdU), and (**D**) zidovudine (AZT). In (**i**) average plots were used to model growth at doses of the analogue and calculate half-maximal growth-inhibition (IC50) doses for each strain. In (**ii**) he average IC50 value in µM is shown for each checkpoint mutant strain. Strains that were not sensitive have an IC50 value above 500 µM (>500, charts), which was the highest dose used. A 95% confidence interval and R^2^ value from 4 experimental replicates were calculated around the IC50 dose. In (**iii**), the area under the curve (AUC) was calculated from a minimum of 4 experimental replicates and is plotted with standard deviation. One-way ANOVA was used to compare AUC for each strain in each analogue (* *p* < 0.05). (**iv**) Cultures were pinned onto the growth medium after 48 h exposure to the analogue. Plates were grown and photographed to detect whether an analogue caused growth arrest or cell death.

**Figure 3 cimb-47-00756-f003:**
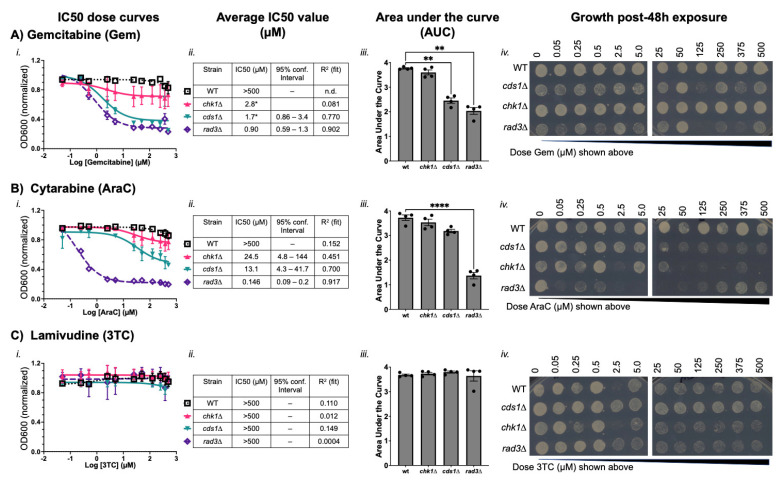
Cytidine analogues gemcitabine and AraC decrease *S. pombe* proliferation. As in Figure 2, cytidine analogues were tested in wild type (WT), *chk1*∆, *cds1*∆, and *rad3∆* strains. (**A**) Gemcitabine (Gem), (**B**) cytarabine (AraC), and (**C**) lamivudine (3TC) were assessed in a minimum of 4 experimental replicates of dose–response curves. An average curve of each strain/drug is presented in panel (**i**). Standard deviation is shown around each average point. The calculated curve was used to generate an IC50 value in µM; strains that did not show growth inhibition by OD600 were described as having an IC50 above 500 µM (**ii**). The area under the curve (AUC), * *p* < 0.05, (**iii**) describes differences in OD600 response curves. AUC values from 4 experimental replicates were plotted and are shown with standard deviation. One-way ANOVA was used to detect differences in AUC (** *p* < 0.01; **** *p* < 0.0001). Absence of connecting line and asterisk indicates that there was no significant difference. In (**iv**), cells were spotted after the 48 h drug incubation onto solid media, and grown in the absence of the drug for 24 h at 30 °C. Pinned spots describe whether a strain is killed by the nucleoside analogue at a given dose.

**Figure 4 cimb-47-00756-f004:**
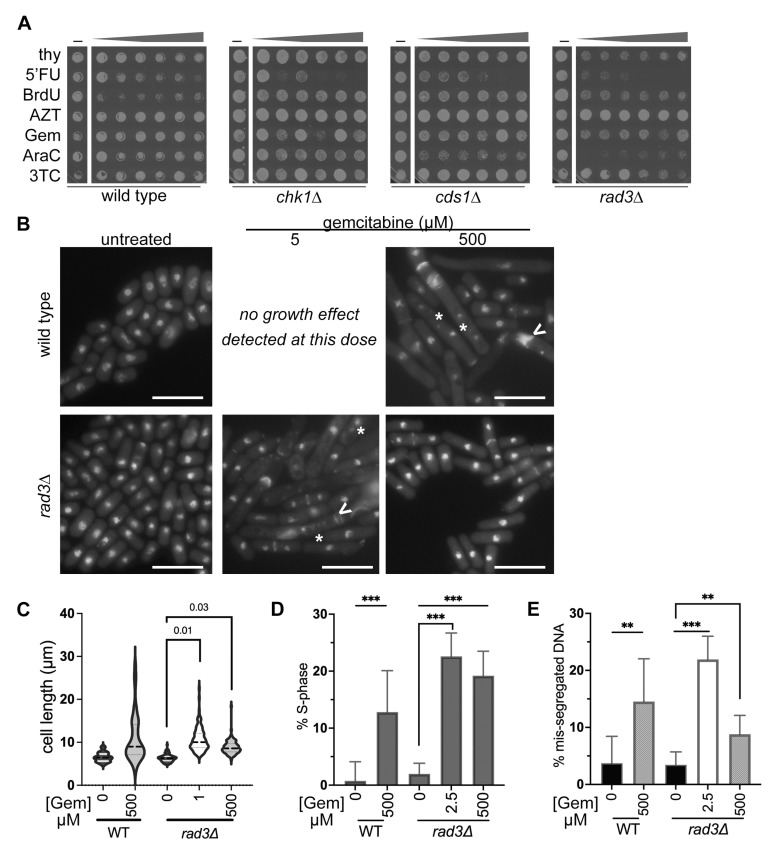
IC50 doses of BrdU and gemcitabine induce DNA mis-segregation, whilst higher doses promote drug-resistant morphologies. (**A**) Cells after 48 h analogue exposure were spotted onto plain YES medium and grown for 48 h at 30 °C. All analogues were grown and compared within the same plate to assess differences caused by each analogue, and different doses. Untreated cells are the left-most column for each strain (wild type, *chk1∆*, *cds1∆*, *rad3*∆); the triangle shows increasing dose from 25 µM dose (left) to 500 µM. We found that strains that are sensitive to analogue and generate an IC50 value (Figure 2 and Figure 3) tended to generate less spot density/cell growth near the IC50 dose. The *cds1∆* and *rad3*∆ strains frequently grew better after high-dose exposure (500 µM) spots, and often generated large single colonies on small background growth (seen in this *cds1*∆ image for AraC and BrdU). (**B**) Wild type and *rad3*∆ cells exposed to gemcitabine were fixed and stained to detect DNA and septum morphologies with no drug, IC50 (*rad3*∆~1 µM), and 500 µM. Wild type cells do not show decreased growth by OD600 and no IC50 sample was possible. Large cells, DNA mis-segregation, and septum abnormalities are seen in *rad3*∆ around the IC50 dose, like what is seen in wild type at a 500 µM dose. Scale bars 10 µm. (**C**) Cell lengths were measured and are presented in different gemcitabine doses from 3 experimental replicates. Repeated measures ANOVA was used to compare cell lengths, and calculated statistical *p*-values are shown above brackets. (**D**) Binucleate (G1) and septated (S-phase) cells were counted as a proportion of the total population from BrdU treated images. A Z-test with Yates’ continuity correction was used to compare each set of untreated (no drug ND) to treated (*** *p* < 0.001). Gemcitabine causes G1/S accumulation at both doses in *rad3*∆, and at 500 µM in wild type. (**E**) Gemcitabine causes mis-segregation in *rad3*∆ cells at the IC50 dose, or in wild type at 500 µM. Cells with anucleate, multiple nuclei (*), or mis-segregated DNA (<) are presented as a fraction of the total number of cells. A Z-test with Yates’ continuity correction was used to compare within each set, ** *p* < 0.01, *** *p* < 0.001.

**Figure 5 cimb-47-00756-f005:**
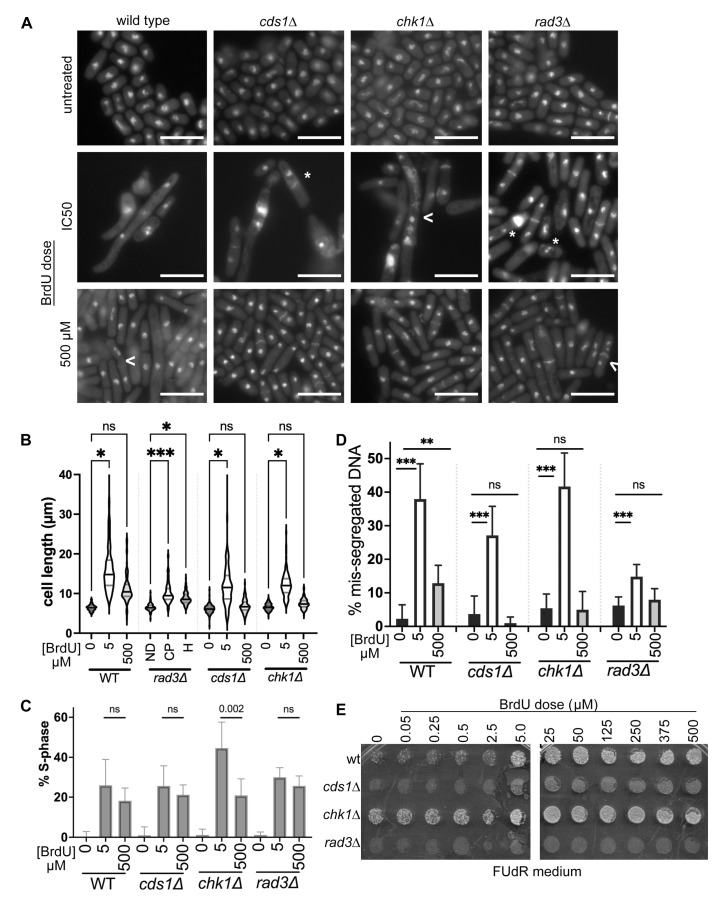
IC50 doses of BrdU cause elongation, G1/S accumulation, and DNA mis-segregation that alter nucleoside analogue resistance. (**A**) Microscopic images after 48 h BrdU treatment, scale bars 10 µm. DNA mis-segregation events including additional nuclei (*) or septation errors (<) were observed in all cells at the IC50 dose of 5–25 µM BrdU or 500 µM. (**B**) Tip-to-tip cell length increases in BrdU and is longest at the IC50 dose. Three (3) experimental replicates were assessed using a violin plot showing the mean and 25%/75% quartiles (note: population averages and SD reported in Table 4). The difference between no BrdU and treatments in each strain was calculated using repeated measures ANOVA with correction (* *p* < 0.05; *** *p* < 0.001, ns—not significant). (**C**) Binucleate (G1) and septated (S-phase) cells were counted as a proportion of the total cell population. A Z-test with Yates’ continuity correction was used to compare each set. Each strain has a statistically significant increase in G1/S-% with both BrdU doses, compared to no BrdU. In *chk1*∆, the proportion of G1/S cells is increased at the 5 µM IC50 dose compared to 500 µM. (**D**) Nuclear mis-segregation is increased in BrdU at an IC50 dose (5 µM) in all strains. Cells with atypical DNA segregation, including anucleate, *cut*, multiple nuclei, or lagging chromosomes, were counted as a proportion of the total number of cells. A Z-test with Yates’ continuity correction was used to compare within each strain, ns—not significant, ** *p* < 0.01, *** *p* < 0.001. Wild type cells show more mis-segregation at 500 µM compared to no BrdU. All other strains did not have a statistically different proportion of mis-segregation at 500 µM. (**E**) After 48 h BrdU exposure, cultures were spotted onto FUdR to test for *hsv-tk*^+^ function. Nucleoside analogue-sensitive cells require *hsv-tk*^+^ for sensitivity to BrdU (0 µM dose column). With increasing BrdU, spots acquire additional FUdR-insensitive cells, even as the number of viable cells decreases (compare to Figure 2(Civ)).

**Figure 6 cimb-47-00756-f006:**
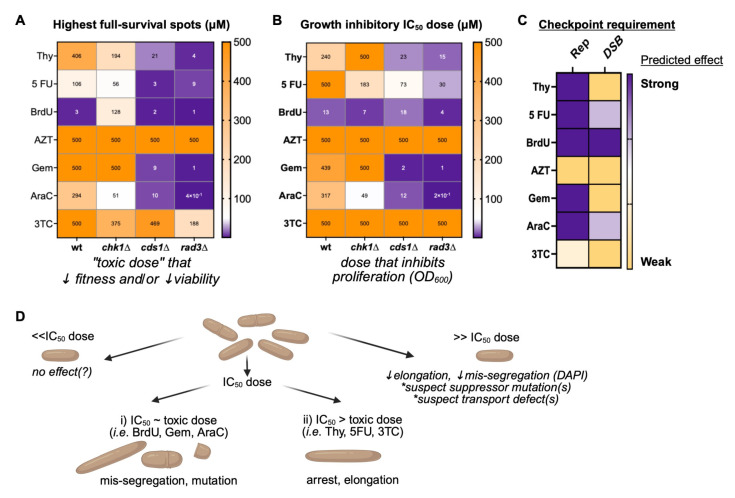
Summary of mechanisms and effects from nucleoside analogue dose and cellular checkpoint status. (**A**). The highest dose of fully surviving spots on YES pinned plates was recorded (n = 4) as the highest dose of each analogue that did not decrease subsequent spot growth on YES. This “toxic dose” was the point above which growth of surviving cells was compromised. (**B**). The IC50 dose inhibited proliferation in the OD_600_ assay. Replicate IC50 values (n = 4) were compared in a heat map. Compounds without an effect or a calculated IC_50_ above 500 were reported as 500. (**C**). Summary of the checkpoint-dependent importance of the replication checkpoint (Rep) or DNA damage checkpoint (DSB) importance on nucleoside analogue sensitivities. A low-dose IC_50_ or toxic dose (<50 µM) affecting *cds1*∆ or *rad3*∆ mutants was scored as Rep “Strong” (score of 3, scale 0 to 3); if *chk1*∆ was impacted then DSB was scored. (**D**). A model of nucleoside analogue dose-dependent effects. Below a strain specific IC_50_ dose of each analogue, no effect is seen. Yet, the wide-ranging of nucleoside analogues suggests that absence of effect may not indicate no effect at all, symbolized by the question mark (?). At the IC_50_ dose, some analogues caused toxicity, mis-segregation, and likely mutation (**i**), including BrdU, gemcitabine, and AraC. A second group of analogues ((**ii**) Thy, 5FU, 3TC) had an IC_50_ dose that was higher than the dose which caused toxicity; we predict that these analogues promote cell elongation and cell cycle arrest. There is a potential for mutation caused by skewed dNTP pools. At high doses above the IC_50_ (right), nucleoside analogues may suppress outwardly toxic effects. Cells do not elongate or show DNA mis-segregation. Predictions are denoted by *. We predict that high-dose nucleoside analogues impact nucleoside transporter functions and promote suppressor mutations that could cause drug resistance. Created in BioRender. Sabatinos, S. (2025) https://BioRender.com/c4gjg0s (accessed on 10 August 2025).

**Table 1 cimb-47-00756-t001:** Fission yeast strains.

Strain	ID	Genotype	Source [Ref]
wild type (WT)	SY 17	*h^+^ leu1–32::hENT1-leu1^+^ (pJAH29) his7–366::hsv-tk-his7^+^ (pJAH31) ura4-D18 ade6-M210*	Susan Forsburg [38]
*cds1∆*	SY 91	*h^+^ cds1Δ::ura4^+^ leu1–32::hENT1-leu^+^ (pJAH29) his7–366::hsv-tk-his7^+^ (pJAH31) ura4-D18 ade6-M210*	Susan Forsburg [39]
*chk1∆*	SY 92	*h^+^ chk1Δ::ura4^+^ leu1–32::hENT1-leu1^+^ (pJAH29) his7–366::hsv-tk-his7^+^ (pJAH31) ura4-D18 ade6-M210*	Susan Forsburg [39]
*rad3∆*	SY 93	*h^+^ rad3∆::ura4^+^ leu1–32::hENT1-leu1^+^ (pJAH29) his7–366::hsv-tk-his7^+^ (pJAH31) ura4-D18 ade6-M210*	Susan Forsburg [39]

**Table 2 cimb-47-00756-t002:** IC50 values calculated from independent replicates, with standard deviation.

	Calculated IC50 ± SD (n)				
Strain	Thy	5FU	BrdU	Gem	AraC
wild type (WT)	>500 ^a^ (5)	383 ± 458 (5)	11.9 ± 4.2 (6)	>500 (4)	>500 (4)
*chk1∆*	>500 ^a^ (4)	>500 (4)	7.7 ± 2.7 (4)	>500 (4)	37.9 ± 35.8 (4)
*cds1∆*	21.6 ± 5.8 (3)	73.6 ± 30.6 (4)	15.9 ± 7.1 (4)	2.2 ± 0.4 (4)	13.6 ± 9.3 (3)
*rad3∆*	14.9 ± 1.1 (4)	36.1 ± 16.2 (4)	3.6 ± 1.3 (4)	1.1 ± 0.1 (4)	0.15 ± 0.05 (4)

In contrast to aggregate IC50 values presented in Figure 2 and Figure 3, this average is from non-normalized data. ^a^: No IC50 value was calculated, and the IC50 value is reported as greater than the 500 µM highest dose used.

**Table 3 cimb-47-00756-t003:** Cells are longest in gemcitabine concentration near the IC50 dose for *rad3∆*.

Gem	Gem Dose (µM)	Mean Length ± SD (µm)	Number of Cells (n)	*p*-Value (Untreated Compared to Treated)
Wild type	0	6.5 (1.1)	100	–
	500	10.9 (5.5)	105	0.005
*rad3∆*	0	6.3 (0.1)	301 (3)	–
	IC50	10.7 (0.7)	312 (3)	0.01
	500	8.8 (0.5)	304 (3)	0.03

The *rad3∆* cells are affected by either 1 µm or 500 µM gemcitabine. Despite no obvious gemcitabine sensitivity, wild type cells are longer at 500 µM gemcitabine dose. Tip-to-tip or tip-to-septum mean cell lengths are shown in µm ± standard deviation. The *p* value indicates if there is a statistically significant difference between the cell lengths at each dose of drug relative to no treatment. A repeated measures one-way ANOVA test was used, assuming Gaussian distribution and the Geisser–Greenhouse correction for matched data sets. Tukey’s multiple comparisons correction was applied.

**Table 4 cimb-47-00756-t004:** All strains elongate in BrdU at an IC50 dose.

BrdU	BrdU Dose (µM)	Mean Length (±SD, µm)	Number of Cells (n)	*p* Value (Significance)
Wild type	0	6.5 (0.9)	289	–
	5	15.9 (1.9)	317	0.05 (*)
	500	11.1 (1.1)	282	0.10 (ns)
*cds1∆*	0	6.3 (0.4)	282	–
	5	12.5 (2.1)	224	0.05 (*)
	500	6.9 (0.3)	208	0.12 (ns)
*chk1∆*	0	6.7 (0.5)	287	–
	5	12.3 (0.7)	269	0.02 (*)
	500	7.6 (1.3)	306	0.60 (ns)
*rad3∆*	0	6.5 (0.4)	306	–
	5	10.1 (0.3)	292	<0.0001 (***)
	500	8.7 (0.3)	308	<0.04 (*)

Cell lengths were measured without BrdU (0 µM), in an IC50 dose (used 5 µM as an average of IC50 values), or in 500 µM BrdU. Cells were measured tip-to-tip or septum to tip. Three (3) experimental populations were measured, and N used in statistics was 3; the total number of cells within these replicates is reported (n). Average cell length per strain is compared between no treatment and each dose. A repeated measure one-way ANOVA test was used with Tukey’s correction. Each *p*-value indicates whether the difference between untreated or BrdU treatment is significantly different (* *p* < 0.01; *** *p* < 0.001; not significant, ns).

## Data Availability

Data supporting reported results are available upon request to Sarah Sabatinos (ssabatinos@torontomu.ca).

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
