# Peer review of "Checkpoint-Dependent Sensitivities to Nucleoside Analogues Uncover Specific Patterns of Genomic Instability"

_cimb, 2025, doi:10.3390/cimb47090756_

Round 1

Reviewer 1 Report

Comments and Suggestions for Authors

Dear Authors, This is a timely manuscript and will benefit Yeast DNA replication & repair experts and beyond emphasising DDR genes mutations & use of nucleoside analogues.

I do not have comments on the data presented as it is well controlled, I would like to know why HU is not used in this context as it also depletes the nucleotide pools.

Second, it is well established in the literature in yeast that checkpoint loss increases fork reversal and increases the ssDNA gaps. These phenotypes (Fork reversal and ssDNA gaps) are exclusively seen in the checkpoint deficient background (Hu et al., Cell, 2012; Lai & Foiani, Cell, 2012), it is important to briefly discuss in the introduction and in the discussion how this will also contribute for the chromosome mis seggregation. I feel it could be nice if the authors also can comment on the essentiality of Homologous Recombination pathway panel profiling, which could help treatment choice to overcome chemo-resistant.

Author Response

Response to Reviewer 1 Comments

  1. Summary

We thank Reviewer #1 for their time and thoughtfulness in reviewing our manuscript. We were delighted to read the comment that our manuscript is “timely” and that it will benefit others in Replication/Repair/Recombination in considering nucleoside analogue use. Please find our detailed responses below and the corresponding revisions/corrections highlighted/in track changes in the re-submitted files.

  1. Reviewer’s general evaluation of manuscript: We thank the reviewer for their time and their assessment that all areas are sufficient for clarity, description, and appropriateness. We address specific comments in section 3.

  1. Point-by-point response to Comments and Suggestions for Authors

Comments 1: “I do not have comments on the data presented as it is well controlled, I would like to know why HU is not used in this context as it also depletes the nucleotide pools.”

Response 1: Thank you for these positive comments on data quality, and this interesting point on alternative compounds with similar effects. We agree with this comment; HU depletes dNTP pools and is an interesting comparator. We did not include HU as an example in the manuscript because nucleoside analogues affect both dNTP pools as well as nascent DNA synthesis directly.

However, it is a point well taken- how does sensitivity to HU compare in the mutant strains we describe, and how is this related to various nucleoside analogue sensitivities? To address this question, we have tested HU in the strains under the same media and growth conditions: PMG medium with supplements, 48 h growth, then pinned onto YES plates. We have included this data as a new supplementary figure (now Figure S2) so that readers can compare HU effects in the mutant strains with the various nucleoside analogues.  The results/discussion of p15 now reads as follows:

“We conclude that lower-dose BrdU promotes DNA replication arrest and DNA damage induction (Figure 6). Generally, we found that an IC50 BrdU dose is often more toxic to cells (Figure 6A, 6B, 6D). These nucleoside analogue effects are distinctive from those caused by hydroxyurea (HU; Supplementary Figure S2A). HU depletes dNTP levels and causes DNA replication arrest through the S-phase checkpoints. Consequently, the IC50 values of HU are lowest in rad3∆, cds1∆ (0.74 µM and 1.1 µM respectively in PMG medium), followed by chk1∆ (7.6 µM) and highest for wild type (14.6 µM). We saw that medium choice affects HU sensitivity (Supplementary Figure S2B). These HU results indicate that gemcitabine, BrdU, and AraC sensitivities in our system were more HU-like and replication-stress dependent. However, it is difficult to deconvolve effects of the analogues on ribonucleotide reductase inhibition versus direct polymerase inhibition, such as the masked chain termination of gemcitabine [71] or DNA damage induction of either BrdU or gemcitabine [72–74].”

Comments 2: Second, it is well established in the literature in yeast that checkpoint loss increases fork reversal and increases the ssDNA gaps. These phenotypes (Fork reversal and ssDNA gaps) are exclusively seen in the checkpoint deficient background (Hu et al., Cell, 2012; Lai & Foiani, Cell, 2012), it is important to briefly discuss in the introduction and in the discussion how this will also contribute for the chromosome mis segregation. I feel it could be nice if the authors also can comment on the essentiality of Homologous Recombination pathway panel profiling, which could help treatment choice to overcome chemo-resistant.

Response 2: We thank the reviewer for these points and agree with their implementation. Concepts of fork reversal and ssDNA gaps are now described in the introduction:

“Replication instability phenotypes including single stranded DNA accumulation, ssDNA gaps, and replication fork reversal, can cause DSBs and damage checkpoint effects in replication checkpoint mutants [48–50].”p2 line 74-76

A comment on homologous recombination was added to the introduction:

“Homologous recombination becomes an important downstream checkpoint response to DSBs and also DNA replication structures [45–47].” p2 line 68-70

Homolgous recombination is discussed in the Conclusions section (p 17), where we mention synthetic lethality testing and Olaparib:

“Homologous recombination pathway testing in this system may help to minimize onset of mutations and drug resistance. This is similar to the strategic lethality principle of the drug Olaparib e.g.[79–81]. Olaparib inhibits polyADP ribose polymerase (PARP), blocking repair and enhancing cell death in BRCA1 or BRCA2 deficient tumours [78–81]. ”

We have included a manuscript version with changes tracked so that the modifications can be seen.

  1. Response to Comments on the Quality of English Language

       No comments were listed

  1. Additional clarifications

We thank Reviewer 1 and have implemented all suggestions.

Reviewer 2 Report

Comments and Suggestions for Authors

It is a very good idea to look at the interplay between checkpoint blockade and different nucleoside analogues used in cancer treatments and/or for antiviral therapies. In some ways this study may have been better done in human/cancer cells. However, there are certainly advantages to doing initial studies in yeast where checkpoint mutants are more readily available and understood. Using S. pombe instead of S. cerevisiae was a good choice and the use of TK and hENT1 stable cell lines was important to make this study feasible.

I was considering whether this manuscript would be easier to follow if rather than going through each drug in turn, that the authors had presented it based on each category of analysis and then go through each compound. I believe it would be very helpful for the authors to create a table that evaluates each compound based on each of the cellular responses: sensitivity to growth/turbidity – wt, and for each mutant, growth post exposure – wt and for each of the mutants, mitotic defects – wt and for each of the mutants, etc.  Because of the widely varying responses to these nucleosides the conclusions from the paper are complex and anything to consolidate and simply the analysis and conclusions would be helpful.

Cds1 is analogous to Chk2 – this should be noted in Figure 1. Likewise, in the Conclusions the authors make clear that Cds-1 represents the DNA replication checkpoint, and Chk1 represents the DNA (DSB?) repair checkpoint – since these terms are used throughout the manuscript it would be helpful for readers to have this clearly indicated in Figure 1.  

In the introduction within the paragraph on DSBs it should be mentioned that DSBs can also caused by replication fork collision with topoisomerase-DNA adducts caused by topoisomerase poisons. These poisons are another common class of anti-cancer agents. Also, the role of 5-FU in RNA synthesis is not discussed. As this may be the primary role of 5-FU action, this needs to be considered in the manuscript.

Could the “suppressor colonies” resulting from 5-FU, AraC, gemcitabine or BrdU treatment retain their clonal resistance to drug, and could this be evident in a OD600 and spot plate test in the next generation? This is suggested by the authors’ discussion and should be addressed.

In a couple of places the authors argue that combinations of nucleoside analogues with ATR inhibition are cautioned. Is this also the case with decreased levels of nucleoside analogues in conjunction with ATR inhibition? This should be considered and discussed, particularly as some clinical studies are now investigating checkpoint inhibition with decreased levels of chemotherapeutics.

More mutational and survival after treatment analyses, such as those in Fig 5E and Fig S1 will definitely be needed for more definitive knowledge of what is occurring in the various types of treated cells. However, this is a huge endeavor and beyond the scope of this paper.

Author Response

Response to Reviewer 2 Comments

  1. Summary

We were delighted by Reveiwer #2’s comments and interest in the manuscript from a fundamental point of view: “It is a very good idea to look at the interplay between checkpoint blockade and different nucleoside analogues used in cancer treatments and/or for antiviral therapies”.

We have dealt with all specific comments to improve the manuscript in section 3 below. However, we were happy to accommodate the suggestions of Reviewer #2 in the manuscript. All revisions/corrections are highlighted using “track changes” in the re-submitted manuscript. We hope that these modifications will not only improve the manuscript’s flow, but will promote our work’s application to different fields and models. Thank you for your time and thoughtful review.

  1. Reviewer’s general evaluation of manuscript: The Reviewer’s point about human and cancer cells is well taken; certainly, this is an active area of investigation in our laboratory. Our results will be specifically tested in these human models in the future, particularly as related to any Rad3/ATR potential effects. We appreciate their understanding of S. pombe as the model organism of choice, and their understanding of the importance of the TK and hENT1 cell lines used.

  1. Point-by-point response to Comments and Suggestions for Authors

 Comments 1: “I was considering whether this manuscript would be easier to follow if rather than going through each drug in turn, that the authors had presented it based on each category of analysis and then go through each compound. I believe it would be very helpful for the authors to create a table that evaluates each compound based on each of the cellular responses... Because of the widely varying responses to these nucleosides the conclusions from the paper are complex and anything to consolidate and simply the analysis and conclusions would be helpful.”

Response 1: We agree- the data may be difficult to consolidate as it spans not only different nucleoside analogues, but also different genotypes linked to 2 cell cycle checkpoints. We have assembled a new figure (Figure 6A) which shows the highest dose of analogue for each strain that did not compromise growth. This allows comparison of cytotoxicity versus cytostatic effects from the drug incubation step. Figure 6B is a second heat map that compares the IC50 values of each analogue in each strain. From these aggregated data, we used a 4 point scale to grade the combination of survival and IC50 effects in each mutant strain relative to wild type. The rad3∆ and cds1∆ cells with impacted growth and low IC50 values show nucleoside analogues with predicted strong replication checkpoint dependency. The rad3∆ and chk1∆ effects on growth and low IC50 values show nucleoside analogues with predicted strong DNA damage and DSB checkpoint dependency.

Below the heat maps, Figure 6D shows a model of the effect of dose on potential outcome. This part of Figure 6 deals with cell length and abnormal mitoses data from BrdU and gemcitabine exposure. Figure 6D also predicts how high-level exposure may affect checkpoint-deficient cells with potentially strong effect outcomes.

Comments 2: “Cds1 is analogous to Chk2 – this should be noted in Figure 1. Likewise, in the Conclusions the authors make clear that Cds-1 represents the DNA replication checkpoint, and Chk1 represents the DNA (DSB?) repair checkpoint – since these terms are used throughout the manuscript it would be helpful for readers to have this clearly indicated in Figure 1.”

Response 2: Thank you for these detailed comments. We have implemented the changes in Figure 1.

Comments 3: “In the introduction within the paragraph on DSBs it should be mentioned that DSBs can also caused by replication fork collision with topoisomerase-DNA adducts caused by topoisomerase poisons. These poisons are another common class of anti-cancer agents. Also, the role of 5-FU in RNA synthesis is not discussed. As this may be the primary role of 5-FU action, this needs to be considered in the manuscript.” 

Response 3: These are excellent points that we have implemented. First, we discuss topoisomerase adducts as sources of DSBs in the introduction, p2 lines 66-69:

“A clinically useful class of DBS drugs includes camptothecin and irinotecan, molecules that bind topoisomerase to the DNA causing DNA replication fork collisions and DSBs in the next S-phase [47–49].”

The reviewer’s point about the different major mechanism of 5-FU on RNA is excellent; this also consolidates some of our findings with respect to the minor but important chk1∆ sensitivity to 5FU. We now describe 5FU more fully in the introduction, p1 38-39:

“5FU also affects mRNA synthesis causing cell death [5–8].”

We also discuss the impact of 5FU on experimental results on p6, lines 247-254:

“These data indicate that 5FU effects require the DNA replication checkpoint to preserve viability. However, even replication-checkpoint competent cells are sensitive to high doses of 5FU, which impairs cell growth. This may reflect that the primary mechanism of 5FU is reportedly upon RNA synthesis [5–8]; consequently, all our mutants tested would be susceptible to 5FU effects. While chk1∆ growth is similar to wild type after 5FU, the subtle difference in IC50 and maximum unaffected doses in spots (Figure 6A, 6B) argues that  impact of DNA damage checkpoint loss should be investigated for differences in mutation kind and profile (i.e. [76]). ”

Comments 4 & 5: “Could the “suppressor colonies” resulting from 5-FU, AraC, gemcitabine or BrdU treatment retain their clonal resistance to drug, and could this be evident in a OD600 and spot plate test in the next generation? This is suggested by the authors’ discussion and should be addressed.”

“In a couple of places the authors argue that combinations of nucleoside analogues with ATR inhibition are cautioned. Is this also the case with decreased levels of nucleoside analogues in conjunction with ATR inhibition? This should be considered and discussed, particularly as some clinical studies are now investigating checkpoint inhibition with decreased levels of chemotherapeutics.

Response 4 & 5: We agree that potential suppressor mutations may be stable and may decrease secondary exposure to drugs. While ATR inhibition may allow lower doses of nucleoside analogue, our data indicates that genome instability present in the ATR-inhibited state may increase the risk of mutation. While a full experimental description of this hypothesis is outside the scope of this current manuscript, we thank the reviewer for their suggestion of essential next steps. We have defined the hypothesis in the conclusion, and link to the absence of clear mutagenic mechanisms and drug resistance on p 17, 614-626:

“Our method can be applied to find a specific IC50 dose that reduces cell growth (Figures 2, 3; Table 2), while minimizing the risk of failure or resistance above the IC50. This could be part of a path a path to uncover new genetic factors that increase nucleoside analogue efficacy. Testing for suppressor mutations that develop at the IC50 dose in checkpoint mutant cells (i.e. cds1∆, rad3∆) may improve treatments and predict causes of drug resistance. To this end, the emergence of suppressor-like colonies in 5-FU, BrdU, AraC, or gemcitabine treatment may reflect stable isolates that are less sensitive to subsequent doses of drug. However, we see off-target effects in rad3∆ above the IC50 dose and loss of FUdR sensitivity. Because ATR is the homologue of S. pombe Rad3, the use of ATR inhibitors to increase nucleoside analogue effects (i.e. [90, 91]) could confer a risk of mutation and later drug resistance. ATR inhibition may allow lower clinical drug doses; future work will determine whether these lower doses are due to enhanced cell killing or if there is a higher risk of analogue-resistance.”

Comments 6: “More mutational and survival after treatment analyses, such as those in Fig 5E and Fig S1 will definitely be needed for more definitive knowledge of what is occurring in the various types of treated cells. However, this is a huge endeavor and beyond the scope of this paper.”

Response 6: We thank Reviewer #2 for their insight and agree with their assessment. Although the mutational/survival assessment is beyond the scope of this paper, the development of resistance caused by genome instability is an essential topic for use with these drugs. As a first pass, our work shows that different loss-of-function checkpoint mutations may influence mutational outcome, and hence risk. The Sabatinos lab looks forward to developing these observations and cataloguing mutation type and frequency, and we have recently begun the analyses of this topic.

  1. Response to Comments on the Quality of English Language

No comments on English language quality were listed.

  1. Additional clarifications

We thank Reviewer 2 for their helpful comments and queries. We have implemented all suggestions, and we can confirm that the next round of future work into types and frequencies of mutations is in progress for a future publication.             
